# Fabrication and Characterization of Fast-Dissolving Films Containing Escitalopram/Quetiapine for the Treatment of Major Depressive Disorder

**DOI:** 10.3390/pharmaceutics13060891

**Published:** 2021-06-16

**Authors:** Manal E. Alkahtani, Alhassan H. Aodah, Omar A. Abu Asab, Abdul W. Basit, Mine Orlu, Essam A. Tawfik

**Affiliations:** 1UCL School of Pharmacy, University College London, 29-39 Brunswick Square, London WC1N 1AX, UK; manal.alkahtani.17@ucl.ac.uk (M.E.A.); a.basit@ucl.ac.uk (A.W.B.); m.orlu@ucl.ac.uk (M.O.); 2Department of Pharmaceutics, College of Pharmacy, Prince Sattam bin Abdulaziz University, Alkharj 11942, Saudi Arabia; 3National Center for Pharmaceutical Technology, Life Science & Environment Research Institute, King Abdulaziz City for Science and Technology, 6086, Riyadh 11442, Saudi Arabia; aaodah@kacst.edu.sa (A.H.A.); oabuasab@kacst.edu.sa (O.A.A.A.)

**Keywords:** major depressive disorder, treatment-resistant depression, coaxial fibers, electrospinning, fast-dissolving films, escitalopram, quetiapine

## Abstract

Major depressive disorder (MMD) is a leading cause of disability worldwide. Approximately one-third of patients with MDD fail to achieve response or remission leading to treatment-resistant depression (TRD). One of the psychopharmacological strategies to overcome TRD is using a combination of an antipsychotic as an augmenting agent with selective serotonin reuptake inhibitors (SSRIs). Among which, an atypical antipsychotic, quetiapine (QUE), and an SSRI, escitalopram (ESC), were formulated as a fixed-dose combination as a fast-dissolving film by coaxial electrospinning. The resultant fiber’s morphology was studied. SEM images showed that the drug-loaded fibers were smooth, un-beaded, and non-porous with a fiber diameter of 0.9 ± 0.1 µm, while the TEM images illustrated the distinctive layers of the core and shell, confirming the successful preparation of these fibers. Differential scanning calorimetry (DSC) and X-ray diffraction (XRD) studies confirmed that both drugs were amorphously distributed within the drug-loaded fibers. The drug-loaded fibers exhibited a disintegration time of 2 s, which accelerated the release of both drugs (50% after 5 min) making it an attractive formulation for oral mucosal delivery. The ex vivo permeability study demonstrated that QUE was permeated through the buccal membrane, but not ESC that might be hindered by the buccal epithelium and the intercellular lipids. Overall, the developed coaxial fibers could be a potential buccal dosage form that could be attributed to higher acceptability and adherence among vulnerable patients, particularly mentally ill patients.

## 1. Introduction

Major depressive disorder (MDD), commonly known as clinical depression, is a seriously debilitating mental disorder. It is one of the most common psychological diseases, affecting more than 264 million people worldwide [1]. The manifold etiology of depression involves genetic, biological, and psychosocial factors. It has been described that the pathophysiological mechanisms of the disease include monoamine deficiency (depletion of the neurotransmitters serotonin, norepinephrine, and dopamine in the central nervous system) and the hyperactivity of the hypothalamic–pituitary (HPA) axis [2]. The pharmacological approach is effective for the management of the disease; there are different classes of antidepressants among which selective serotonin reuptake inhibitors (SSRIs) are considered as the first-line treatment approach [3].

Approximately one-third of MDD patients fail to achieve remission (i.e., absence of symptoms and full return to functioning as prior to disease onset) or response to first-line antidepressant treatments, which is widely known as treatment-resistant depression (TRD) [4]. Patients diagnosed with TRD are projected to have a more severe course of illness and are at a higher risk of suicide. The psychopharmacological treatment strategies for TRD include switching antidepressants within a class or between classes, combining two or more different antidepressants, or augmenting treatment by adding a non-antidepressant therapy to an active therapy [5,6].

Escitalopram (ESC) is an SSRI used for the treatment of MDD and anxiety disorder. It exerts its antidepressant activity by selectively binding to the human serotonin transporter (SERT) and inhibiting serotonin (5-HT) reuptake, leading to elevated serotonin levels in synaptic clefts [7]. Due to its favorable tolerability, high selectivity, and preventive effect on relapse and recurrence of MDD, ESC shows clinically relevant superiority over other SSRIs [8].

Augmenting agents to SSRIs include atypical antipsychotics, among which quetiapine (QUE) is approved by the United States Food and Drug Administration (USFDA) for this purpose [9]. QUE is primarily approved for the treatment of schizophrenia and bipolar disorders. It acts by antagonizing dopamine D1 and D2, adrenergic a1, histaminergic H1, and serotonergic 5-HT2 receptors [10]. Nothdurfter et al. compared the antidepressant activity of ESC/QUE combined with ESC monotherapy and reported a significant inhibitory effect of the combined therapy on HPA system activity [11]. QUE is considered as a BCS (i.e., biopharmaceutics classification system) class II drug that has low solubility and undergoes extensive hepatic metabolism [12].

Fast-dissolving films (FDFs), also known as fast-disintegrating films, are solid dosage forms that consist of a thin hydrophilic polymer sheet that dissolves and releases the drug within a few seconds when placed in the oral cavity without the need for chewing or water intake [13,14]. It is one of the novel approaches that increase patient’s acceptability and usability owing to its ease of administration and rapid disintegration [13,14,15,16]. This fast disintegration property can be a particular advantage in the reformulation of antidepressants, since dysphagia is a serious and common issue in patients with mental illness [17]. Multiple studies have reported oral mucosal (buccal and sublingual) absorption of FDFs which enhance drug bioavailability via bypassing the first-pass metabolism, and avoid pre-systemic clearance [18,19,20,21,22,23].

FDFs could be formulated using techniques such as electrospinning, solvent casting, 3D printing, ink-jet printing, and hot-melt extrusion. Each technique offers advantages and has limitations. For example, 3D printing (3DP) is an alternative fabrication method for polymeric oral film preparation that is able to produce personalized medicines on demand [24,25]. However, thermally labile APIs may not be the best candidate for most 3D printing techniques and hot-melt extrusion where high temperatures are required [26]. In such case, other methods such as electrospinning and solvent casting might be implied. Thermal inkjet (TIJ) printing has been successfully used to fabricate FDFs; Vuddanda et al. (2018) and Alomari et al. (2018) were able to produce FDFs that were personalized and contained a drug combination using TIJ [27,28]. Films’ characteristics differ depending on the used formulation technique. It was previously reported that electrospun films have enhanced mechanical properties such as flexibility, durability, and plasticity over solvent-casted ones [29]. In addition, ink-jet-printed films have better mechanical properties and more physical stability compared to those formulated by the solvent-casting method [30].

Electrospinning is a process that generates micro- to nano-sized fibers, which can be achieved by employing electrostatic force to the drug–polymer solution. This technique allows the formation of an amorphous solid dispersion that enhances drug solubility and dissolution, making it particularly attractive for BCS class II drugs [31]. Other advantages of electrospinning include ease of fabrication, compatibility with thermolabile materials, construction of a variety of structures, and relatively low start-up cost. The ability to co-deliver multiple drugs and control their release rate can be achieved through the use of coaxial or multiaxial electrospinning techniques [32,33]. There is a wide range of potential applications of electrospun fibers in the medical field which include, but are not limited to, tissue engineering, wound dressing, antibacterial dressing, biological sensing, and drug delivery [34].

Therefore, the current study aims to formulate a fixed-dose combination of ESC and QUE in coaxial electrospun fibers as an alternative dosage form to the marketed tablets of both drugs for the management of MDD, in order to achieve high patient acceptability, compliance, and drug regimen adherence. This is by accelerating the onset of action and avoiding the first-pass metabolism through oral mucosal delivery, which in turn will reduce the clinical doses of each drug and, hence, their unpleasant adverse effects.

## 2. Materials and Methods

### 2.1. Materials

Polyvinylpyrrolidone (PVP; Mw = 1300 kDa), ethyl alcohol (≥99.5%), sodium chloride NaCl (≥99.5%), sodium phosphate dibasic Na_2_HPO_4_ (≥99.0%), potassium chloride KCl (99.0–100.5%), potassium phosphate monobasic KH_2_PO_4_ (≥99.0%), and hydrochloric acid HCl (36.5–38.0%) were purchased from Sigma-Aldrich (St. Louis, MO, USA), HPLC grade acetonitrile was obtained from PanReac AppliChem ITW Reagents (Barcelona, Spain), and escitalopram oxalate (ESC) (≥99.95%) and quetiapine fumarate (QUE) (≥99.13%) were kindly donated by Saudi Pharmaceutical Industries and Medical Appliances Corporation (SPIMACO), also known as SPIMACO ADDWAEIH (Riyadh, Saudi Arabia). Deionized water was generated by Milli Q, Millipore (Billerica, MA, USA), and was used in the preparation of simulated salivary fluid (SSF) pH 6.8 and phosphate-buffered saline (PBS) pH 7.4. All organic solvents were of high-performance liquid chromatography (HPLC) grade.

### 2.2. Preparation of Drug-Loaded Fibers Using Electrospinning Technique

The drug-loaded spinning solution was prepared by dissolving PVP in ethanol at a concentration of 10% (*w*/*v*) and stirred for at least 120 min over a magnetic stirrer at ambient temperature until complete dissolution, according to the modified method of Illangakoon et al. [35]. Both ESC and QUE were added at a concentration of 1% (*w*/*v*) to separate PVP polymer solutions (i.e., separately in two different vials) and stirred for a further 60 min to obtain homogenous solutions. For the blank (i.e., drug-free) spinning solution, only the PVP solution was prepared similarly to the abovementioned method, but with no addition of any drug.

The preparation of the coaxial drug-loaded fibers was optimized using an electrospinning setup purchased from Spraybase^®^ (Dublin, Ireland). A coaxial needle with inner and outer diameters of 0.45 and 0.9 mm, respectively, was connected to two different syringes that contained the drug-loaded spinning solutions, in which ESC was in the outer layer and QUE in the inner layer, respectively. The distance between the spinneret and the collector, which was covered with aluminum foil, was adjusted to 15 cm, whereas the flow rate was maintained at 0.5 mL/h for each layer. The applied voltage was between 7 and 9 kV. The entire process was carried out under ambient conditions (20–25 °C and relative humidity of 30–45%). The blank fibers were prepared using similar conditions to the drug-loaded fibers, but with no addition of drugs.

### 2.3. Scanning Electron Microscopy (SEM)

The morphological characterization of the fibers’ surfaces was evaluated by SEM (JSM-6010 PLUS/LA, JEOL Inc., Peabody, MA, USA). Fibers were collected directly onto aluminum foil. The samples were coated with platinum (10 nm) in a JEC-3000FC auto fine coater (JEOL Inc., Peabody, MA, USA) to make them electronically conductive. After coating, the samples were transferred and imaged under an excitation voltage of 5 kV. Fiber size analysis was performed by measuring the diameter of at least 100 fibers using ImageJ software (National Institute of Health, Bethesda, MD, USA).

### 2.4. Transmission Electron Microscopy (TEM)

The distinguishing of the inner and the outer layers of the coaxial fibers was assessed by TEM (JEM-1010, JEOL Inc., Peabody, MA, USA). Fibers were collected directly on a copper grid during electrospinning. The prepared grids were imaged at an accelerating voltage of 80 kV.

### 2.5. Thermal Analysis and Physical-State Characterization

#### 2.5.1. Differential Scanning Calorimetry (DSC)

The DSC analysis of blank and drug-loaded fibers, ESC, QUE, PVP, and their physical mixture (PM) at a similar drug-to-polymer ratio to the drug-loaded fibers, was performed using Mettler Toledo DSC 3+ Star System (Columbus, OH, USA). Aliquot samples of each group (weight 5 ± 0.5 mg) were placed in an aluminum pan sealed with pin-holed aluminum lids. The samples were equilibrated at 0 °C for 30 min then were heated to 200 °C at a rate of 10 °C/min under a nitrogen flow of 20 mL/min. The test was performed in triplicate and the data were analyzed using the STAR^e^ software provided by Mettler Toledo (Columbus, OH, USA).

#### 2.5.2. X-ray Diffraction (XRD)

The solid-state materials characterization was studied by MiniFlex 600 benchtop diffractometer (RigaKu, Tokyo, Japan) on blank and drug-loaded fibers, ESC, QUE, PVP, and their PM. The XRD was supplied with Cu Kα radiation (λ = 1.5148 227 Å) at a voltage of 40 kV and a current of 15 mA. The samples were scanned in triplicate by fixing them on glass holders and the data were recorded over the 2θ range between 2° and 50° at a scan speed of 5°/min. XRD patterns were plotted and analyzed using OriginPro^®^ 2021 software (OriginLab Corporation, Northampton, MA, USA).

#### 2.5.3. Fourier-Transform Infrared Spectroscopy (FTIR)

The intermolecular compatibilities and interactions between components were analyzed using a Thermo Scientific™ Nicolet™ iS20 FTIR Spectrometer (Waltham, MA, USA) on blank and drug-loaded fibers, ESC, QUE, PVP, and their PM. The samples were scanned, in triplicate, over the range of 4000–550 cm^−1^, with the spectral resolution set at 4 cm^−1^. An average of 32 scans per sample were recorded. Background scans were performed in all experiments. Spectra were plotted and analyzed using OriginPro^®^ 2021 software (OriginLab Corporation, Northampton, MA, USA).

### 2.6. Disintegration Test of the Electrospun Fibers

The disintegration of both blank and drug-loaded fibers was assessed following the method described in Tawfik et al. (2021) [13]. A 2 × 2 cm square piece of each fiber mat was dropped into a petri-dish containing 6 mL of pre-warmed SSF (pH 6.8) under gentle stirring until complete detachment. The experiment was carried out in a thermostatic shaking incubator (Excella E24 Incubator Shaker Series, New Brunswick Scientific Co., Enfield, CT, USA) at 37 °C. The results represent the mean ± SD of at least three replicates.

### 2.7. High-Performance Liquid Chromatography (HPLC) for Drugs Determination and Quantification

An HPLC method that separates and quantifies ESC and QUE was developed using modified methods of [36,37,38,39]. The analysis was performed in a Waters e2695 HPLC system that consisted of a Waters^®^ 717 plus autosampler, Waters 600 binary pump, and Waters 2489 UV/detector (Waters Technologies Corporation, Milford, MA, USA). The chromatographic separation was achieved using isocratic elution and an Xbridge C_18_ column (250 × 4.6 mm, 5.0 µm) with a temperature that was kept at 40 °C. The mobile phase was composed of 0.1 M ammonium acetate buffer adjusted with 1.0 mL triethylamine as solvent A and acetonitrile as solvent B in a ratio of 50:50 (*v*/*v*). The flow rate was maintained at 0.65 mL/min, the injection volume was set at 25 µL, and the analysis was carried out at a detection wavelength of 235 nm. Both drugs were dissolved in simulated saliva fluid (SSF) that was prepared according to Marques et al. (2011) by mixing 8 g NaCl, 0.19 g KH_2_PO_4_, and 2.38 g Na_2_HPO_4_ in 1 L of distilled water, and the pH was adjusted to 6.8 using 5M HCL [36].

### 2.8. Determination of the Drug Loading (DL), Entrapment Efficiency (EE%), and Fiber Yield (Y) of the Drug-Loaded Fibers

To determine the DL and EE%, at least three square pieces of the drug-loaded fibers, with weights of 20 ± 0.5 mg, were dissolved in 10 mL SSF and were kept at ambient temperature for six hours to guarantee complete dissolution of the fibers. The samples were evaluated using the developed HPLC method, and the DL and EE% were calculated using the following equations, respectively:(1)$$\mathrm{DL}=\frac{\mathrm{Entrapped}\;\mathrm{drug}\;\mathrm{amount}}{\mathrm{Yield}\;\mathrm{of}\;\mathrm{fibers}\;\mathrm{amount}\;}$$
(2)$$\;\mathrm{EE}\%=\frac{\mathrm{Actual}\;\mathrm{drug}\;\mathrm{amount}\;}{\mathrm{Theoretical}\;\mathrm{drug}\;\mathrm{amount}\;}\times 100$$

The Y of the drug-loaded fibers was calculated by the following equation:(3)$$Y\%=\frac{\mathrm{Actual}\;\mathrm{amount}\;\mathrm{of}\;\mathrm{fibers}}{\mathrm{Theoretical}\;\mathrm{amount}\;\mathrm{of}\;\mathrm{fibers}}\times 100$$
where the theoretical amount of fibers was calculated using the amount of the solid materials (polymer and drug) in the total volume of spinning solution that was spun.

All the results of the DL, EE%, and Y are represented as the mean ± SD of at least three replicates. More details on calculating the DL and EE% are presented in the Appendix A.

### 2.9. Determination of the In Vitro Drug Release of the Drug-Loaded Fibers

Since the developed formulation can be considered as an FDF, the standard Pharmacopeia (United State or British Pharmacopeias) dissolution methods do not accurately mimic the condition [35]. Therefore, a previously developed experimental release study was employed according to [35]. Certain weights of the drug-loaded fibers that were approximate to 30 ± 0.5 mg were placed in glass vials and sunk using custom-made sinkers in prefilled (15 mL) and pre-warmed SSF (pH 6.8) as the release medium. The release study was carried out in a thermostatic shaking incubator (Excella E24 Incubator Shaker Series, New Brunswick Scientific Co.) at 37 °C and 100 RPM. One milliliter samples were withdrawn from the solutions at predetermined time points and replaced with an equal volume of fresh pre-warmed SSF to maintain the total volume of 15 mL. The amount of drug released was determined by the developed HPLC method. The cumulative release percentages were measured as a function of time and were calculated according to the following equation:(4)$$\mathrm{Cumulative}\;\mathrm{release}\%=\frac{\mathrm{Cumulative}\;\mathrm{drug}\;\mathrm{amount}}{\mathrm{Theoretical}\;\mathrm{drug}\;\mathrm{amount}}\times 100$$

The findings represent the mean ± SD of at least three replicates.

### 2.10. Ex Vivo Permeation Study of the Drug-Loaded Fibers

To determine whether the developed formulation enhanced the permeability of ESC and QUE, an ex vivo permeability study was conducted using the Franz diffusion cell system (PermeGear, Hellertown, PA, USA). Freshly removed bovine buccal membranes were collected from a local abattoir and placed immediately in ice-cold phosphate-buffered saline (PBS) pH 7.4, as recommended by [40]. PBS was prepared by mixing 8 g NaCl, 0.2 g KCl, 0.24 g KH_2_PO_4_, and 1.44 g Na_2_HPO_4_ in 1 L of distilled water, and the pH was adjusted to 7.4 using 5M HCL [41].

A 1 ± 0.3 mm-thick membrane was carefully excised using surgical scissors and fine-point forceps and measured with an electronic digital Vernier caliper. The buccal mucosa was mounted between the donor and receptor compartments of the Franz diffusion cell with the epithelium facing the donor and the connective tissue facing the receptor then the compartments were clamped together. Simulated salivary fluid SSF (pH 6.8) and phosphate-buffered saline PBS (pH 7.4) were used to fill the donor and receptor compartments, respectively. The system was maintained at 37 ± 0.02 °C to mimic the temperature of an in vivo environment, and the diffusion medium was stirred at 600 RPM. The open ends of the apparatus were covered with parafilm to prevent evaporation. The system was left to equilibrate for 15 min, then the donor fluid was removed and replaced with a pre-warmed SSF containing approximately 1 mg of each drug either as a powder or an equivalent amount of the formulation. At predetermined time points up to 4 h, a 250 µL sample was withdrawn from the receptor compartment and replaced by the fresh pre-warmed buffer to maintain the experimental conditions. The withdrawn samples were analyzed by the developed HPLC method. The data were statistically analyzed using OriginPro^®^ 2021 software (OriginLab Corporation, Northampton, MA, USA) and the results represent the mean ± SD of three replicates.

### 2.11. Stability Studies

To investigate the storage stability of the physical form of the drug-loaded fibers, these fibers were kept at ambient conditions (20–25 °C and relative humidity 30–45%) for 4 months. The solid-state characteristics of the fibers were evaluated using DSC and XRD as previously described in Section 2.5.1 and Section 2.5.2, respectively.

### 2.12. Statistical Analysis

The disintegration, DL, EE%, Y, in vitro, and ex vivo experiments were all performed as three independent replicates and the data are reported as mean value ± SD. OriginPro^®^ 2021 software (OriginLab Corporation, Northampton, MA, USA) was used to statistically analyze the data. One-way ANOVA was used to compare the means of the ex vivo permeation study, with the *p*-value set as 0.05.

## 3. Results and Discussion

### 3.1. Fiber Morphology and Size Analysis

The blank (i.e., drug-free) and drug-loaded coaxial fibers were successfully prepared. Their morphology and fiber diameter distribution were evaluated by SEM, as illustrated in Figure 1. The fibers demonstrated smooth, cylindrical, and non-porous surfaces with continuous and bead-free morphology and exhibited an excellent uniformity in terms of diameter distribution. The average diameters of the blank and drug-loaded fibers were 1 ± 0.2 and 0.9 ± 0.1 µm, respectively. The diameter of the PVP blank fibers was consistent with previously reported studies [42,43]. This clearly indicates high-quality fibers that have been achieved by both the materials’ and processing parameters’ optimization.

Drug-loaded coaxial fibers were also observed by TEM to evaluate the inner structure and to distinguish between both the inner and outer layers. Figure 2 shows clear distinctive core/shell layers that represent the inner (PVP and QUE) and outer (PVP and ESC) layers. The TEM images also indicate that the drug-loaded core/shell fibers were successfully fabricated. It appeared that the thickness of the inner layer was considered large; this is probably due to the similar flow rate that was used for both layers and the miscibility between the systems, which was also observed in the poly (lactic-co-glycolic) acid (PLGA)/PVP coaxial system of [44]. In this study, it was proposed to use a coaxial fibers system, to avoid loading the fibers with a high concentration of drugs (2% *w*/*v* that is equally divided for each drug) into single-layered fibers that would reduce the drug-to-polymer ratio, preventing the encapsulation of the drugs, hence their presence on the fibers’ outer surface [32]. The successful preparation of this coaxial system was satisfying to further test the fibers using thermal and solid-state analyses, as well as in vitro and ex vivo release tests.

### 3.2. Differential Scanning Calorimetry (DSC)

DSC was used to determine the physical form of the coaxial drug-loaded fibers. As illustrated in Figure 3, both pure drug powders ESC and QUE are in crystalline form, due to the presence of sharp endothermic melting peaks at 154.3 and 176.8 °C, respectively. The peaks are in agreement with the melting points that are stated in the literature, i.e., 153.8 °C for ESC [45], and 175.8 °C for QUE [46].

Pure PVP showed a broad endotherm between 44.7 and 115.9 °C, which indicates water evaporation, owing to the hygroscopic nature of this polymer [32]. The polymer is also known to be amorphous; therefore, it is expected to show no sharp melting endotherms [47]. In the PM thermogram, the drug’s characteristic endothermic peaks that represent their melting points were merged, broadened, and shifted toward lower temperatures (142.4 and 166 °C for ESC and QUE, respectively), and their intensity was reduced. This might be due to the high polymer concentration and drugs being uniformly distributed in the PM, leading to total miscibility of the molten drugs in the polymer [48].

The blank fibers showed a broad endothermic peak between 50.5 and 110.3 °C, which is water evaporation. The DSC thermogram of the drug-loaded fibers was similar to the blank fibers, with water evaporating between 54.4 and 107.1 °C. There were no melting peaks related to the drugs, suggesting the molecular dispersion transformation of the drugs due to the electrospinning process, which was reported previously [33,35]. The DSC analysis indicated that the fibers are in amorphous form.

### 3.3. Physical Form Characterization by XRD

XRD was used to investigate the physical state of the drugs after electrospinning, as it was previously reported that this technique can lead to the formation of amorphous solid dispersion [35,44,47]. The results in Figure 4 illustrate that the pure ESC and QUE drugs were in crystalline form, which was identified by the presence of multiple distinct sharp peaks (Bragg reflections) in their corresponding XRD patterns. In the ESC diffractogram, the diffraction peaks at 13.14°, 13.16°, 16.06°, 19.2°, 20.94°, and 27.12° were observed, which are in agreement with the ESC XRD patterns of [49]. QUE diffraction patterns showed distinct peaks at 14.64°, 15.9°, 16.74°, 19.68°, 20.0°, and 23.0° that are similar to those of the QUE diffractogram reported by [50].

PVP is clearly amorphous; its diffractogram exhibited a broad halo and lacked any characteristic reflections. Peaks of both ESC and QUE were observed in the PM at diffraction angles of 19.68°, 20.94°, and 27.12°. Despite the weak intensities, the presence of these peaks indicated that the drugs were in crystalline form within the PM. Both blank and drug-loaded coaxial fibers appeared as amorphous, since there were no distinctive diffractions that could be observed in their patterns and their appearance as broad halos. For the drug-loaded fibers, the lack of distinctive peaks of the drugs could be explained by the molecular dispersion of these drugs within the fibers that were transformed by the electrospinning process. This expected observation is consistent with [33,35,44,47]. Overall, this XRD finding further confirmed the results of the DSC, which suggest that the drug-loaded coaxial fibers were in the amorphous solid dispersion form.

### 3.4. Physical Form Characterization by FTIR

FTIR was used to investigate the presence of any intermolecular interactions between the components of the drug-loaded fibers. Drug–polymer compatibility is an essential element to prevent phase separation and to ensure the stability of the resultant formulation. Molecular interaction between components could be indicated by shifts in peak positions in the IR spectra. The FTIR spectra of ESC, QUE, PVP, PM, blank, and drug-loaded fibers are shown in Figure 5.

The characteristic spectrum of ESC that was obtained by FTIR exhibited peaks at 2954 (C–H stretching), 2231 (C≡N stretching vibration), and 1221 cm^−1^ (C–N stretching). This spectrum of ESC is consistent with the ESC FTIR spectrum of [51,52]. The pure QUE spectrum showed distinctive peaks at 3310 (O–H stretching), 2944 (C–H stretching), 1597 (N–H bending), and 1063 cm^−1^ (C–O stretching). This spectrum is in agreement with the QUE IR spectrum of [53,54].

The characteristic peaks of the pure PVP at 1656, 1420, and 1284 cm^−1^ corresponded to the stretching vibrations of the chemical groups C=O, C–H, and C–N, respectively. The spectrum showed two broad bands at 3650–3100 and at 3030–2790 cm^−1^ that were related to the O–H stretching, from adsorbed water, and the C–H stretching, respectively. This spectrum is consistent with the PVP IR spectrum of [33,35,44,47,55]. The stretching peaks of PVP at 1656 cm^−1^ and between 3650 and 3100 cm^−1^ were present in the PM, blank, and drug-loaded fibers IR spectra. As illustrated in the PM spectrum, the characteristic peaks at 1220 and 2231 cm^−1^ confirmed the presence of ESC in the PM, while peaks at 3313, 2945, 1598, and 1064 cm^−1^ were related to QUE.

The intermolecular interactions, such as the formation of hydrogen bonding between the drugs and PVP, were indicated by the slight shift in the peak position from 1656 cm^−1^ in the pure PVP to 1661 cm^−1^ in the drug-loaded fibers spectrum. Such interactions would lead to good compatibility between fiber components and subsequently, might enhance the long-term stability of the drug-loaded fibers [35,47].

### 3.5. Disintegration Time of the Coaxial Fibers

The disintegration time is an important qualitative feature of dosage forms intended to be used in the oral cavity, such as FDFs. Previous studies have explored the use of Petri-dishes for disintegration testing of electrospun fibers and FDFs [35,44,47,56,57,58]. According to the USFDA recommendations, drug products that are administered without chewing or liquids should disintegrate in ≤30 s [59]. Both blank and drug-loaded fiber mats were found to be disintegrated ultra-rapidly in simulated salivary fluid, as shown in Figure 6. Complete disintegration was observed in 2 ± 0 s for drug-loaded fibers, and 2 ± 1 s for blank fibers. This rapid disintegration (≤3 s) of electrospun fibers is in agreement with the previous reports that used similar polymer systems (i.e., PVP) [31,43]. Factors that contributed to such ultrafast disintegration include the use of a water-soluble and fast-dissolving polymer, namely PVP, the high surface-area-to-volume ratio of electrospun fibers, and the amorphous nature of the fibers. Therefore, the drug-loaded coaxial fibers are considered a suitable system for buccal delivery of ESC and QUE.

### 3.6. Drug Loading (DL), Entrapment Efficiency (EE%), and Fiber Yield (Y) of the Drug-Loaded Coaxial Fibers

The DL and encapsulation efficiency (EE%) for the drug-loaded fibers were determined by the developed HPLC method that is shown in the Appendix A and Appendix A. The DL and EE% of ESC and QUE were 22 ± 1 µg/mg and 92 ± 3% for ESC and 21 ± 1 µg/mg and 89 ± 5%, for QUE, respectively. Due to both drugs having similar initial concentrations (0.5% *w*/*v*) and similar core and shell flow rates (0.5 mL/h), the resultant DL and EE were almost equivalent. This high EE% of > 85% was also obtained in the coaxial fiber systems of [40,60].

The Y of the formulated drug-loaded coaxial fibers was 85 ± 1%. The 15% loss in the Y might have resulted during the electrospinning process—for instance, the jet could be dragged away from the collection area—or during the collection process of the fibrous mat in which trace fibers were left unpeeled off the aluminum foil. Nevertheless, this Y can be considered high and will be beneficial when scaling up the system.

### 3.7. In Vitro Drug Release Determination of the Drug-Loaded Coaxial Fibers

An in vitro dissolution test was carried out in SSF at pH 6.8 to mimic the human oral cavity. Both drugs showed relatively rapid dissolution profiles and were released by more than 50% in the first 5 min, as shown in Figure 7. The shell of the fiber contained ESC, whereas the core contained QUE. ESC showed a faster release profile in the initial 10 min which could be explained by two main points. Firstly, the shell of the fiber was in contact with the dissolution medium; thus, it dissolved rapidly. Secondly, the shell had a larger surface area compared to the core that allowed ESC to be released slightly more rapidly than QUE [32]. At around 15 min, the core/shell drug release profiles started to follow a similar pattern. After 30 min, more than 80% of both drugs were released and full drug release was obtained after 120 min.

Different molecular weights of PVP have been used for electrospinning; however, it has been reported that with lower molecular weight (MW) PVP, higher concentrations are required to produce uniform fibers in contrast to high MW PVP [43]. Indeed, the MW of the polymer plays an important role in the dissolution process. In the current study, high MW PVP was used (1300 kDa) which consisted of a very long molecular chain that was extensively entangled and require more time to unravel. Consequently, a slower dissolution rate can be obtained, compared to lower MW PVP. Similar MW PVP was used by Rasekh et al. and Aburayan et al. for preparing indomethacin- and halicin-loaded fibers, respectively; the full drug release for both studies was reported to be over 30 min [55,61].

In general, several factors could enhance the solubility and dissolution rate, including the use of hygroscopic polymers, such as PVP, which accelerate the fibrous mat disintegration, dissolution, and drug release into solution. The high surface-area-to-volume ratio can ensure a high contact area with the dissolution medium, subsequently enhancing drug release. Finally, the drug being in the amorphous form may eliminate the need to overcome any lattice energy barrier to the dissolution, which is created by the intermolecular forces between molecules in the solid state [31].

### 3.8. Ex Vivo Permeation Study of the Drug-Loaded Coaxial Fibers

An ex vivo permeability study using a Franz diffusion cell was conducted to study the permeation ability of the ESC and QUE after being formulated by electrospinning. The results show that QUE permeation was enhanced compared to the control at different time points; however, it was not significantly different (*p* > 0.05). As illustrated in Figure 8, there was a strong correlation between increasing the contact time and enhanced permeation profile which could be due to the time needed for the drug to be completely released from the fibers. This finding was also in agreement with the in vitro drug release result, which shows that the full release of QUE and ESC was achieved after 120 min.

The enhanced QUE permeability was due to the presence of the drug in the amorphous form. In addition, PVP has been shown to act as a permeability enhancer through increasing drug solubility and subsequently, its penetration [61,62]. On the other hand, ESC demonstrated a lack of permeation for up to four hours. Several hypotheses were suggested; however, to investigate the permeability behavior of each drug and to eliminate any possibilities for drug interactions or competition at the receptor site, another experiment using each drug alone was performed. The results supported the finding generated previously; QUE was permeated in a similar fashion to that described in Figure 8, whereas ESC did not permeate.

ESC is suitable for buccal delivery since it has a low MW (324.39 g/mol), low daily dose (10–20 mg), and high log P (0.79) [51]. Although, its permeability might be hindered by the buccal epithelium and intercellular lipids extruded by membrane-coating granules which represent the permeability barrier for buccal penetration of compounds [63]. Several studies have linked these barriers to the limited penetration of compounds. For instance, Shojaei et al. reported that the buccal epithelium can act as a permeability barrier for dideoxycytidine and the addition of the penetration enhancer menthol significantly increased its permeation [64]. Furthermore, the intercellular lipids that ensure the cohesion of epithelial cells have been found to be a rate-limiting step for the diffusion of hydrophilic compound fluorescein isothiocyanate in porcine buccal tissue [65]. Consequently, enhancing the drug buccal permeability could be achieved by incorporating penetration enhancers in the formulation, such as a blend of polymers that contain polyethylene glycol (PEG), i.e., PVP/PEG [63,66].

### 3.9. Stability Studies

The storage stability of the drug-loaded fibers was investigated since amorphous materials tend to convert to a crystalline state over time [32]. A number of studies have used different test conditions and storage periods to examine the long-term stability of electrospun FDFs. Hence, the method used here was adapted and modified from [47,67]. The XRD pattern and DSC thermogram of the aged fibers are shown in Figure 9. The XRD diffractogram (Figure 9a) depicted a broad halo and lacked any characteristic peaks of the drugs, which confirmed that the fibers remained in the amorphous form. The DSC thermogram (Figure 9b) demonstrated the absence of the endothermic peaks that are related to the crystalline drugs, which is in agreement with the XRD finding.

In the current study, PVP (MW of 1300 kDa) was used as a carrier polymer; the inclusion of long-chain polymer molecules in the fibers may hinder the recrystallization [32]. Mohapatra et al. studied the effect of different MWs of PVP on the stability of solid dispersions, and founded that higher MW PVP can increase the stability of the amorphous indomethacin compared to lower MW PVP [68]. PVP electrospun fibers have also shown good long-term stability in previously reports [35,66,69]. Overall, the electrospun fibers retained their stability as amorphous systems with no recrystallization occurring during a storage period of 4 months at ambient conditions (20–25 °C and relative humidity 30–45%).

## 4. Conclusions

Drug-loaded coaxial fibers containing a fixed-dose combination of ESC and QUE for the treatment of MDD were successfully formulated and characterized. The prepared fibers showed smooth, un-beaded, and non-porous surfaces with an average diameter of 0.9 ± 0.1 μm. The DSC and XRD confirmed the molecular dispersion of both drugs within the electrospun fibers, while the FTIR verified a good structural compatibility between the drugs and the polymer. The DL for ESC (22 ± 1 µg/mg) and QUE (21 ± 1 µg/mg) were almost equivalent, owing to the similar initial drug concentration and flow rate that was used in the electrospinning process. The drug-loaded fibers exhibited a disintegration time of 2 s, making them highly suitable for oral cavity administration. The drugs’ release profiles were satisfactory, in which more than 50% of ESC and QUE were released after 5 min, and a full drug release was obtained after 120 min. The ex vivo study on bovine buccal membrane confirmed the permeation of QUE but not ESC, which could be related to the structural integrity of the buccal epithelium. Overall, the development of FDFs could be a promising alternative to the conventional oral dosage forms (i.e., tablets and capsules) of antidepressants for the vulnerable population, especially mentally ill patients. This is due to the ease of administration of electrospun fibers that will contribute to higher drug regimen adherence rates. However, further optimization of the fibers is required to enhance the permeability of ESC, perhaps by incorporating drug permeability enhancers.

## Figures and Tables

**Figure 1 pharmaceutics-13-00891-f001:**
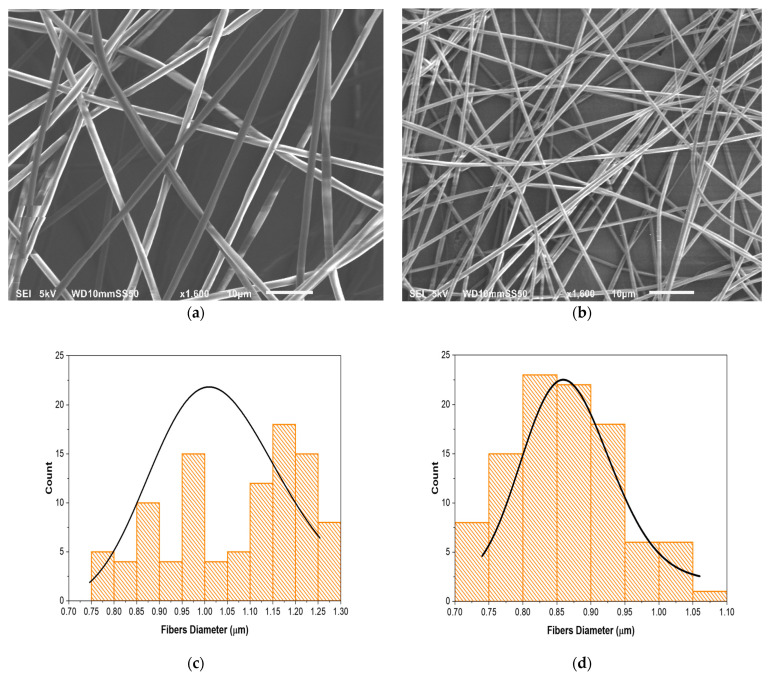
SEM images of (**a**) blank fibers, (**b**) drug-loaded fibers, (**c**) blank fibers size distribution, (**d**) drug-loaded fibers size distribution. The blank and drug-loaded fibers were smooth, un-beaded, and non-porous with a fiber diameter of 1 ± 0.2 and 0.9 ± 0.1 µm, respectively.

**Figure 2 pharmaceutics-13-00891-f002:**
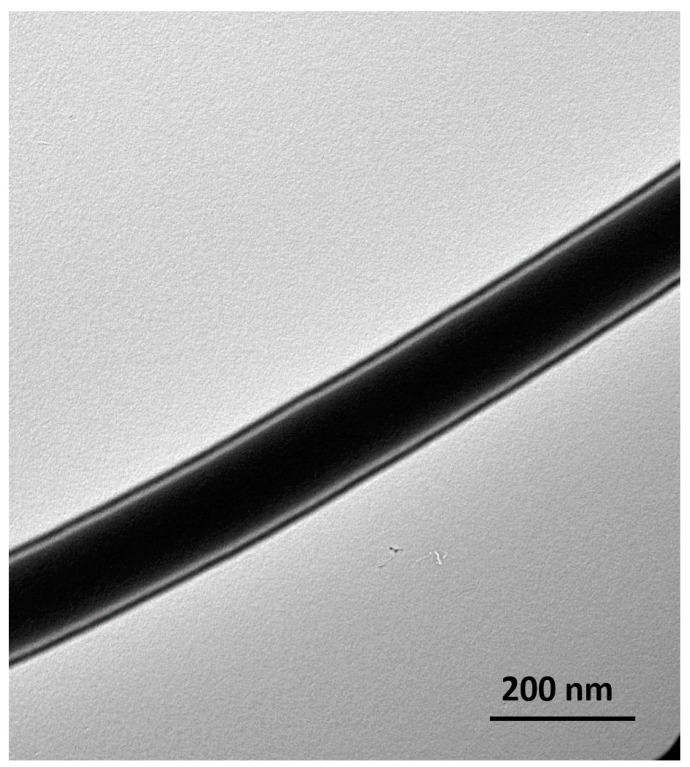
TEM image of drug-loaded coaxial fiber, showing the distinctive fiber layers of the core and shell.

**Figure 3 pharmaceutics-13-00891-f003:**
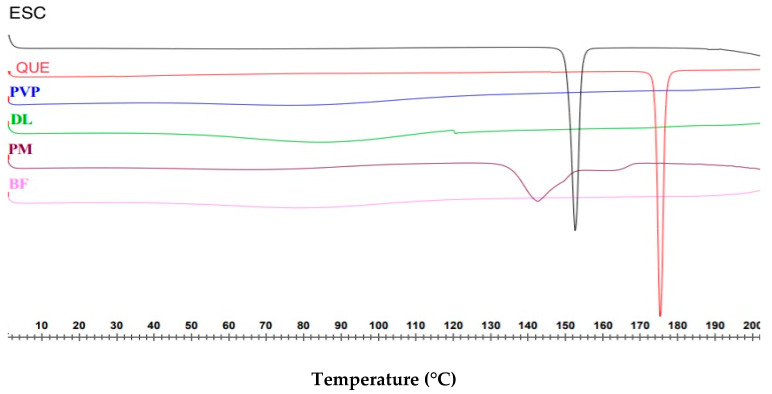
DSC data for ESC, QUE, PVP, physical mixture (PM), drug-loaded fibers (DL), and blank fibers (BF), indicating the melting temperatures of ESC and QUE at 154.3 and 176.8 °C, respectively. DL fibers exhibited no drug endothermic peaks that indicated the molecular dispersion transformation of the drugs within these fibers.

**Figure 4 pharmaceutics-13-00891-f004:**
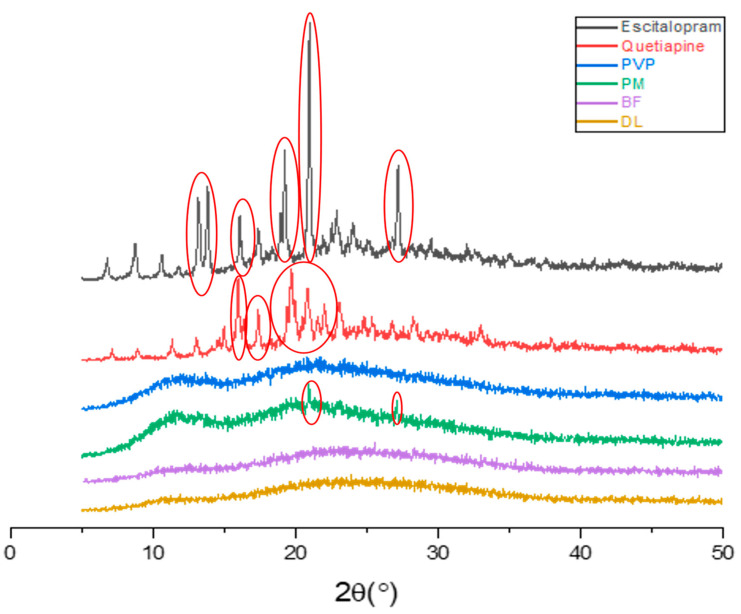
XRD diffraction patterns of ESC, QUE, PVP, physical mixture (PM), blank fibers (BF), and drug-loaded fibers (DL), showing that both drugs have distinctive peaks, indicating that they are in the crystalline form, while the polymer is in the amorphous form represented by broad halos. There are distinctive peaks in the PM representing both drugs which lacked in the BF and DL coaxial fibers (broad halos), indicating the molecular dispersion of both drugs in the DL.

**Figure 5 pharmaceutics-13-00891-f005:**
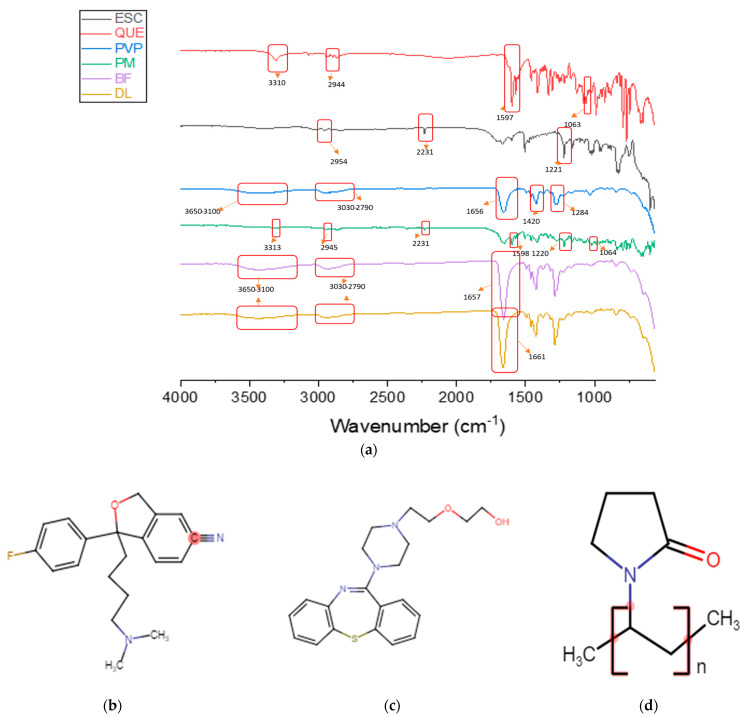
(**a**) FTIR spectra of ESC, QUE, PVP, PM (physical mixture), BF (blank fibers), and DL (drug-loaded fibers), showing each material’s distinctive peaks. Chemical structures of (**b**) ESC, (**c**) QUE, and (**d**) PVP were drawn by chem-space.com.

**Figure 6 pharmaceutics-13-00891-f006:**
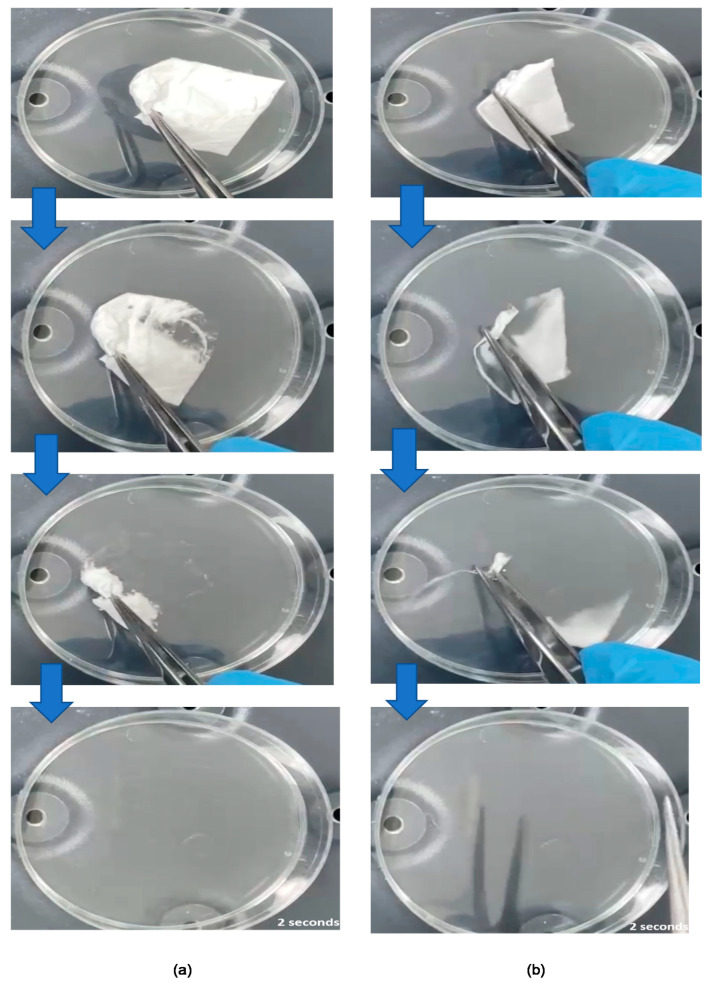
Photos of the disintegration of (**a**) blank fibers, and (**b**) drug-loaded fibers, showing that both fibrous systems have a disintegration time of 2 s.

**Figure 7 pharmaceutics-13-00891-f007:**
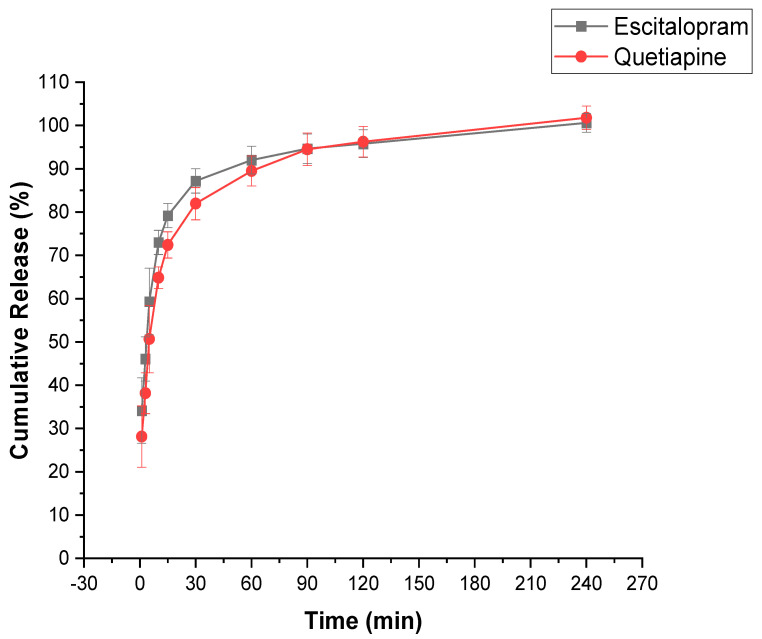
In vitro dissolution profile of the drug-loaded fibers, showing the release of more than 50% of ESC and QUE in the first 5 min and a full release of both drugs after 120 min.

**Figure 8 pharmaceutics-13-00891-f008:**
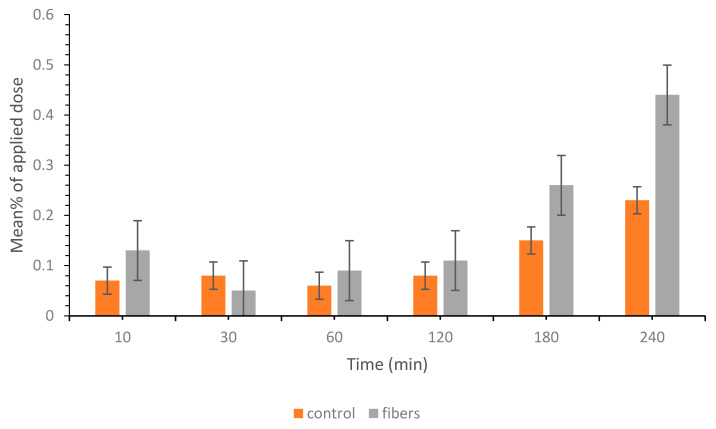
Ex vivo permeation data of QUE in bovine buccal mucosa showing the mean % of applied dose with respect to time. QUE in the fibers had an enhanced permeation starting from 60 min, up to 240 min; however, it was not significantly different (*p* > 0.05).

**Figure 9 pharmaceutics-13-00891-f009:**
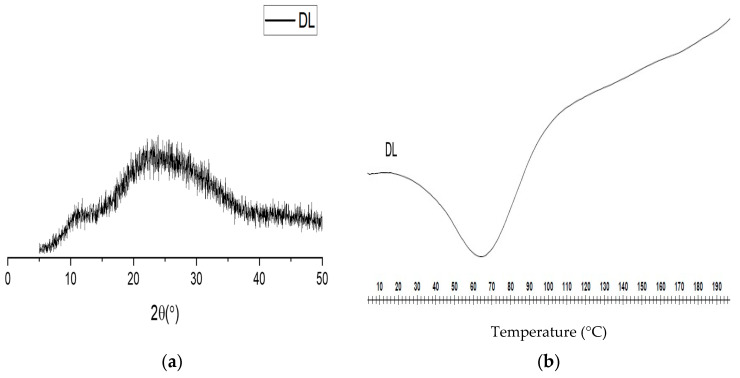
XRD pattern (**a**) and DSC thermogram (**b**) of the drug-loaded fibers (DL), showing a broad halo and lack of drug endothermic peaks, respectively. This indicates the amorphous nature of the DL fibers after 4 month storage period at ambient conditions (20–25 °C and relative humidity 30–45%), which suggests good stability.

## Data Availability

The data presented in this study are available on request from the corresponding author.

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
