# Peer review of "Fabrication and Characterization of Fast-Dissolving Films Containing Escitalopram/Quetiapine for the Treatment of Major Depressive Disorder"

_pharmaceutics, 2021, doi:10.3390/pharmaceutics13060891_

Round 1

Reviewer 1 Report

I read with interest this paper. The authors should consider to 

  • Shorten the introduction and be more focused
  • Correct typos and English spelling
  • Add a figure reassuming all the experiments performed in a flowchart to increase the readability. 
  • Why not using in vivo (on experimental animals, with an adjusted dosage of your formulation) the  permeability/absorption and serum evaluation of the new compound?
  • Add a discussion section including the potential application in clinical practice and designing the subsequent study to assess the real effectiveness of this formulation.
  • Which are the ' further optimization' cited in the conclusions?

Reviewer 2 Report

The technology behind this research is valuable. However, those two drugs they have used to be loaded in combination have significant issues from the clinical perspective.

Some of these issues include the fixed concentration of 1:1 w/v% of the two drugs achieving in an equal loading of the two drugs (QUE = 22 ± 1 μg/mg and ESC 21 ± 1 416

μg/mg). From a technical perspective, this is a great outcome; however, it might not be clinically relevant, and there are low possibilities for translation to the clinic. Questions for authors are: How these doses in this research were selected while different doses of the two drugs are used in the clinic? Examples of some doses used in the clinical trials were quetiapine XR 100 mg/h.s. and increased by 100 mg q h.s. q day to a target dose of 300 mg/h.s. and escitalopram 5 mg/a.m. starting dose; increase to 10 mg/a.m. at week 2 and continued at 10 mg/a.m.

As you can see, clinical doses by weight of the drugs are hugely different from what has used in the current study. Another significant issue is that clinicians need to have flexibility over changing the dose of such medications with applications in mental disorders, and if they are bound to fixed doses, that would not be a favourable option for them. Hence this research has major fundamental issues to go ahead with clinical translation in the future and will probably stay at the level of a publication.

There are some other issues in the writing of the manuscript. Some of the are as follows:

  1. Major depressive disorder (MMD) is a major cause of disability: wording needs revision.
  2. Quetiapine and Escitalopram are generic names and need to be written in small letters.
  3. DSC and XRD need to be defined or removed from abstract
  4. Sentence in line 26-29 is too long and complex. Consider rewriting it.
  5. “ability to co-delivery of multiple drugs” consider revising to … co-deliver multiple drugs and control …
  6. Line 102-106: consider splitting to shorter sentences.
  7. 12. Statistical Analysis: for meaningful in vitro results, experiments need to be performed at least in three independent experiments, each run in triplicate.
  8. Consider rewriting the sentence line 279-281
  9. It is recommended to use void in place of blank for unloaded fibres.

Round 2

Reviewer 1 Report

The authors correctly addressed the point raised before.

Author Response

Thank you so much 

Reviewer 2 Report

Thanks to the authors for their thorough responses.

There is still one point to be clarified. regarding comment 12 in statistical analysis, it is worth to not that when authors mention "in triplicates" means three times that is run simultaneously. However, it is recommended that for in vitro studies 3 independent triplicates to be performed to confirm the repeatability and reproducibility of the results. Some high impact factor journals do not accept any experiment that has been performed only in triplicate if not repeated three times, each independent to another experiment. If authors has performed 3 independent experiments, then they should clarify it. if not, those results are not highly reliable.

Author Response

Thank you for your thoughtful and insightful comment. This is true and we apologized for miswriting this section. Please find an improved version below; 

The disintegration, DL, EE%, Y, in vitro and ex vivo experiments were all performed as three independent replicates and the data were reported as mean value ± SD. OriginPro® 2021 software (OriginLab Corporation, Northampton, MA, USA) was used to statistically analyze the data. One-way ANOVA was used to compare the means of the ex vivo permeation study, with the P-value was set as 0.05.